# Rare disruptive mutations and their contribution to the heritable risk of colorectal cancer

Daniel Chubb[1,*], Peter Broderick[1,*], Sara E. Dobbins[1,*], Matthew Frampton[1], Ben Kinnersley[1], Steven Penegar[1], Amy Price[1], Yussanne P. Ma[1,†], Amy L. Sherborne[1], Claire Palles[2], Maria N. Timofeeva[3], D. Timothy Bishop[4], Malcolm G. Dunlop[3], Ian Tomlinson[2] & Richard S. Houlston[1,5]

Colorectal cancer (CRC) displays a complex pattern of inheritance. It is postulated that much of the missing heritability of CRC is enshrined in high-impact rare alleles, which are mechanistically and clinically important. In this study, we assay the impact of rare germline mutations on CRC, analysing high-coverage exome sequencing data on 1,006 early-onset familial CRC cases and 1,609 healthy controls, with additional sequencing and array data on up to 5,552 cases and 6,792 controls. We identify highly penetrant rare mutations in 16% of familial CRC. Although the majority of these reside in known genes, we identify POT1, POLE2 and MRE11 as candidate CRC genes. We did not identify any coding low-frequency alleles (1–5%) with moderate effect. Our study clarifies the genetic architecture of CRC and probably discounts the existence of further major high-penetrance susceptibility genes, which individually account for >1% of the familial risk. Our results inform future study design and provide a resource for contextualizing the impact of new CRC genes.

[1] Division of Genetics and Epidemiology, The Institute of Cancer Research, London SM2 5NG, UK. [2] Molecular and Population Genetics Laboratory, Wellcome Trust Centre for Human Genetics, University of Oxford, Oxford OX3 7BN, UK. [3] Centre for Population Health Sciences, University of Edinburgh, Edinburgh EH8 9AG, UK. [4] Section of Epidemiology and Biostatistics, Leeds Institute of Cancer and Pathology, University of Leeds, St James's University Hospital, Leeds LS9 7TF, UK. [5] Division of Pathology, The Institute of Cancer Research, London SM2 5NG, UK. * These authors contributed equally to this work. † Present address: Canada's Michael Smith Genome Sciences Centre, BC Cancer Agency, Vancouver, British Columbia, Canada, V5Z 4S6. Correspondence and requests for materials should be addressed to R.S.H. (email: richard.houlston@icr.ac.uk).

Colorectal cancer (CRC), a major cause of cancer-related mortality, displays a complex pattern of inheritance. The genetic architecture of CRC susceptibility encompasses a broad spectrum of risk, from rare, highly penetrant germline mutations associated with well-characterized syndromes to common polymorphisms, each individually conferring small risks. However, much of the familial risk remains unexplained but is widely postulated to be enshrined in unidentified, rare, high-impact variants. This class of susceptibility is mechanistically important and highly relevant to the clinical management of both familial and sporadic CRC[1]. Our recent application of next-generation sequencing to familial cases led to the discovery of mutations in POLE and POLD1 that predispose to CRC, thus providing evidence for the existence of hitherto-unidentified, rare, high/moderate-penetrance susceptibility alleles[2]. We therefore implemented whole-exome sequencing (WES) to quantify the contribution of rare disruptive variants to the heritable risk of CRC and identify new susceptibility genes.

The high baseline rate of rare, neutral germline variants makes the identification of rare CRC predisposition alleles problematic. We therefore designed a study for the extreme phenotype of early-onset CRC, as heritability is substantially greater when diagnosed young and/or is familial. In 2007, we established the UK National Study of Colorectal Cancer Genetics (NSCCG) as a bio-repository for studying susceptibility to CRC[3]. From 23,693 CRC cases (diagnosed <70 years) with European Ancestry recruited between 2007 and 2013, we identified 1,143 with early-onset CRC (≤55 years) documented to have at least one first-degree relative with CRC, a highly enriched subset representing <2% of all CRC. One thousand and twenty-eight of the 1,143 Discovery series of cases were subjected to WES. For comparison, we analysed WES data on 1,644 UK population controls from the 1958 Birth Cohort (1958BC) with no history of malignancy. We identify high-penetrant rare mutations in 16% of familial CRC. Although the majority of these reside in known genes, we identify POT1, POLE2 and MRE11 as candidate CRC genes. Our study clarifies the genetic architecture of CRC and probably discounts the existence of further major high-penetrance susceptibility genes.

## Results

**Whole-exome sequencing.** Cases and controls were sequenced using Illumina TruSeq exon capture and Hi-Seq 2000 technology (Fig. 1 and Methods). To avoid erroneous findings, we performed alignment and variant calling of all samples simultaneously (Methods). We excluded 57 subjects with low-quality data or non-European ancestry (Supplementary Figs 1 and 2). Each captured base was sequenced to an average depth of 48× across samples; cases and controls had similar sequencing metrics (Supplementary Table 1). We estimated sensitivity and specificity of calls using a subset of 1,332 samples, which had been genotyped using Illumina HumanExome-12v1_A Beadchips, identifying high levels of concordance (Methods). The final data set comprised 1,006 Discovery cases and 1,609 controls (Fig. 1 and Supplementary Fig. 1) for which the characteristics are detailed in Supplementary Table 2.

**Analysis of recurrent variants.** We first examined individual, moderately low-frequency coding variants (minor allele frequency (MAF) 1–5%) for an association with CRC risk. No signal deviated from that expected by chance. To maximize the detection of a statistically significant association, we performed a meta-analysis of our Discovery series with Illumina HumanExome-12v1_A Beadchip genotypes on an additional series of 5,552 cases and 6,792 controls (Fig. 1 and Methods).

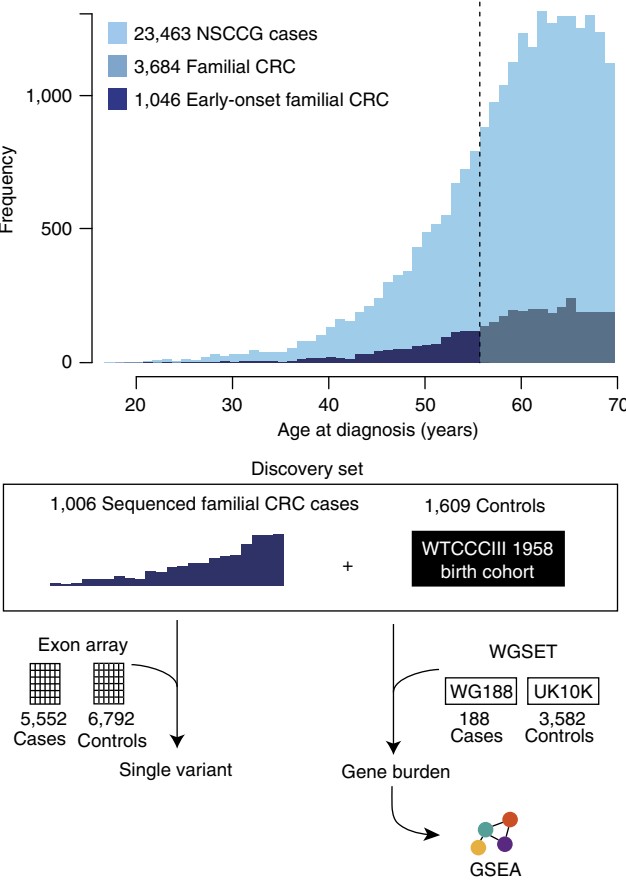

**Figure 1 | Study design to investigate the contribution of rare disruptive mutations to the heritable risk of CRC.** WES was performed on a highly enriched subset of CRC patients recruited to the NSCCG and a series of UK population controls from the 1958 Birth Cohort. After QC, this Discovery set comprised 1,006 early-onset (diagnosed ≤55 years) familial cases (≥1 first-degree relative with CRC) and 1,609 controls. To test the hypothesis that low-frequency variants confer risk for CRC, we performed a meta-analysis alongside 12,344 UK samples genotyped on the Illumina HumanExome-12v1_A Beadchip. To test the hypothesis that a burden of rare mutations in a gene confers risk for CRC, we performed a burden test on rare (<1% frequency) coding variants and subsequent meta-analysis alongside additional sequencing data from 3,770 UK samples (WGSET). Finally, we performed GSEA to investigate whether the burden of rare variants are overrepresented in a specific biological pathway.

Meta-analyses did not identify a statistically significant association for any single variant (that is, $P > 4.0 \times 10^{-7}$).

**Gene-based analysis of rare variants.** We next examined the impact of rare alleles (MAF <1%) collectively within a gene on CRC risk by aggregating single nucleotide variants (SNVs) and indels ('T1' test) in each gene and comparing the counts between cases and controls (Methods). Empirical P-values were obtained using permutation. Acknowledging the limitations of in-silico prediction to enrich for harmful alleles, we considered three sets of variants: Class 1, disruptive mutations (nonsense and frameshift); Class 2, predicted damaging (disruptive plus missense predicted to be damaging and splice donor/acceptor-site): and Class 3, all non-synonymous changes. To account for multiple testing we set the threshold for exome-wide significance at $P = 8.0 \times 10^{-7}$ (Bonferroni correction for 20,000 genes and 3 classes of variants).

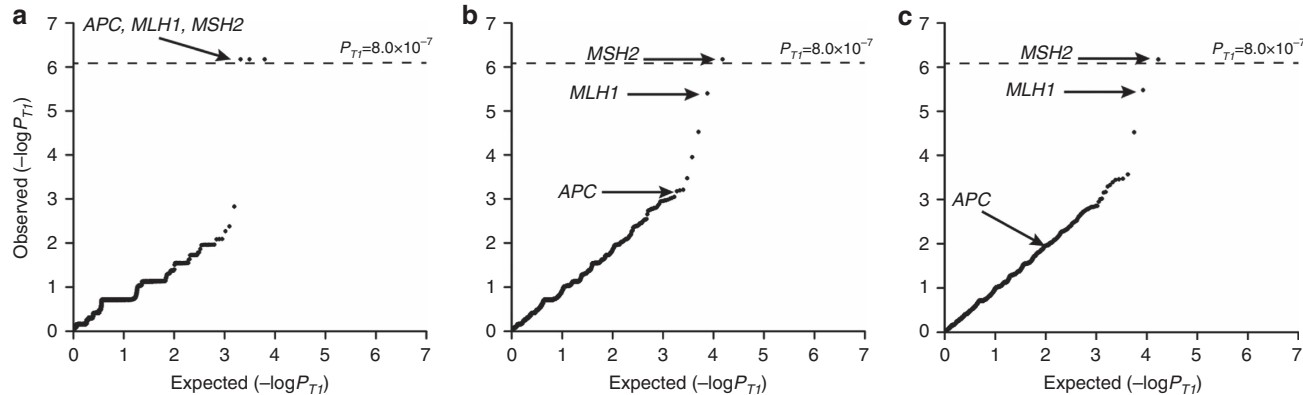

**Figure 2 | Quantile–quantile plot of the T1 burden test results applying three different variant classifications.** (**a**) Class 1, disruptive; (**b**) Class 2, predicted damaging; (**c**) Class 3, all non-synonymous variants. *MLH1*, *MSH2* and *APC* genes annotated. Dotted line corresponds to $P_{T1}$-value of $8.0 \times 10^{-7}$.

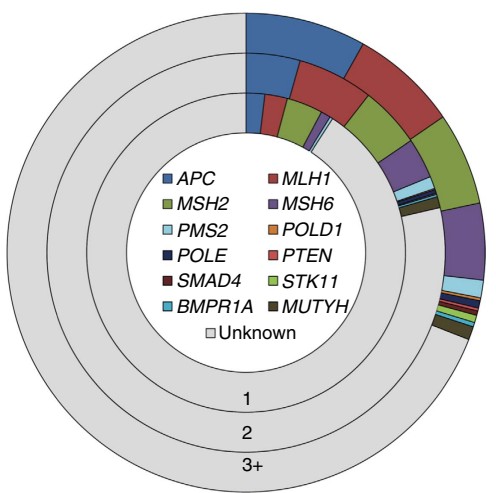

**Figure 3 | Contribution of rare mutations in known predisposition genes to familial CRC applying three different variant classifications.** (1) Class 1, disruptive; (2) Class 2, predicted damaging; (3) Class 3, all non-synonymous variants, and in addition, splice region variants catalogued as pathogenic by InSight or Clinvar.

Class 1 variants showed the strongest enrichment with significant associations being shown for known susceptibility genes: *MSH2*, *MLH1* and *APC* (Fig.2 and Supplementary Data 1). T1 association for *APC* based on Class 2 variants was far less significant, emphasizing the issue of assigning pathogenicity to missense variants (Supplementary Data 2). Moreover, for well-documented CRC genes, including *MSH6* and *PMS2*, irrespective of class, T1 associations did not attain statistical significance, reflecting also their more limited contribution to CRC susceptibility (Fig.2 and Supplementary Data 1–3). None of the cases were identified as being carriers of multiple Class 1 mutations in a known gene (Supplementary Note 1).

To estimate the maximal contribution of known susceptibility genes to familial CRC, we relaxed sequencing filters and manually assessed additional rare variants incorporating ClinVar/InSight annotations (Methods). Nine per cent of the CRC cases could be ascribed to rare Class 1 mutations in the established high-penetrance genes. Even including the contribution of all rare non-synonymous variants, only 31% of familial CRC is explained (Fig.3 and Supplementary Table 3).

To search for novel CRC susceptibility genes we confined analysis to the 863 cases, which did not carry a probable disease-causing mutation in an established high-penetrance CRC gene. Although no T1 association was exome-wide significant (Supplementary Data 1), we sought support for promising associations using whole-genome sequencing (WGS) data on an additional 188 familial cases (WG188) in conjunction with WGS data on 3,582 UK10K controls—WGSET (Fig.1 and Methods). We restricted this analysis to genes with Class 1 nominally significant T1 P-values in the Discovery series. Evaluating candidate genes for support in the WGSET, the presence of recurrent variants and biological plausibility identified three novel potential CRC genes—*IL12RB1*, *LIMK2* and *POLE2* (Tables 1 and 2). The recurrent *IL12RB1* truncating variants p.Gln542Ter (three WES and two WGSET cases) and p.Gln32Ter (one WES and one 1958BC control) cause recessive IL12-Rβ1 deficiency[4]. Interestingly, another *IL12RB1* truncating mutation (Gln376Ter) has been linked with intestinal gastric cancer[5]. In *LIMK2*, we identified the recurrent variant p.Gln574ArgfsTer12 in five WES cases and one 1958BC control. Positive selection for decreased LIMK2 activity during CRC initiation and progression has been documented[6], and of the 18 *LIMK2* mutations catalogued by COSMIC, 14 were identified in colorectal tumours[7]. The recurrent *POLE2* p.Leu469PhefsTer17 variant was detected in three WES and one WG188 cases; the mutation is rare in Europeans (Exome Aggregation Consortium (ExAC) frequency 10/27,173; Table 1 and Supplementary Table 4). POLE2 is a subunit of the polymerase epsilon enzyme complex. *POLE* mutation causes polymerase proofreading-associated polyposis[8]; hence, there is a strong likelihood that *POLE2* mutation will affect CRC. Support for this comes from the recent identification of *POLE2* mutations in two families segregating CRC and polyposis[9].

**Gene-set enrichment analysis.** As far as they have been deciphered, cancer susceptibility genes for the same tumour type are often implicated in the same biological processes. To complement our T1 analysis of single genes, we conducted a gene-set enrichment analysis[10] (GSEA) based on the GO Biological Process ontology using our entire Discovery series to identify genes, which individually may not be so remarkable as to be detectable at the exome-wide level of significance (Methods). The DNA_REPLICATION gene set was significantly associated with CRC (Q-value $<0.1$ in gene-set permutation test; Supplementary Table 5). This enrichment is driven by disruptive mutations in *MLH1*, *MSH2*, *PMS2* and *MSH6*, but also *POLE2*, and *POT1* and *MRE11A* genes hitherto not previously implicated in CRC susceptibility.

*POT1* and *MRE11A* represent credible CRC predisposition genes *a priori*. Both genes participate in telomere length

**Table 1 | Disruptive mutations identified in candidate CRC susceptibility genes with biological relevance.**

| Gene | Mutation | | Exomes | | | WGSET | | | ExAC | $P_{T1\_META}$ |
|---|---|---|---|---|---|---|---|---|---|---|
| | cDNA change | Protein change | Case | Control | $P_{T1}$ | Case | Control | $P_{T1}$ | | |
| IL12RB1 | | | 4 | 1 | 0.029 | 2 | 0 | 0.001 | | 8.7E − 04 |
| | c.94C > T | p.Gln32Ter | 1 | 1 | | 0 | 0 | | 3 | |
| | c.1624C > T | p.Gln542Ter | 3 | 0 | | 2 | 0 | | 5 | |
| LIMK2 | | | 5 | 1 | 0.012 | 2 | 0 | 0.001 | | 2.7E − 04 |
| | c.1711_1712insC | p.Gly574ArgfsTer12 | 5 | 1 | | 0 | 0 | | 151 | |
| | c.1742dupG | p.Cys582LeufsTer4 | 0 | 0 | | 1 | 0 | | 0 | |
| | c.2049_2050insA | p.Gln684ThrfsTer16 | 0 | 0 | | 1 | 0 | | 0 | |
| POLE2 | | | 3 | 0 | 0.021 | 2 | 1 | 0.004 | | 9.5E − 04 |
| | c.1406dupT | p.Leu469PhefsTer17 | 3 | 0 | | 2 | 0 | | 10 | |

n(cases/controls): Exomes(863/1,604), WGSET(188/3,582), ExAC(0/27,173)
CRC, colorectal cancer; ExAC, Exome Aggregation Consortium.
Genes with biological relevance identified in T1 gene burden analysis of disruptive mutations with $P_{T1} < 0.05$ support in the WG set and/or the presence of rare recurrent variants.

**Table 2 | Disruptive mutations identified in novel candidate CRC susceptibility genes.**

| Gene | Mutation | | Exomes | | WGSET | | ExAC |
|---|---|---|---|---|---|---|---|
| | cDNA change | Protein change | Case | Control | Case | Control | |
| MRE11A | | | 3 | 0 | 0 | 0 | |
| | c.1726C > T | p.Arg576Ter | 1 | 0 | 0 | 0 | 4 |
| | c.1066delC | p.His356ThrfsTer34 | 1 | 0 | 0 | 0 | 0 |
| | c.21-6_26del | p.Leu7fsTer18 | 1 | 0 | 0 | 0 | 3 |
| POT1 | | | 2 | 0 | 1 | 0 | |
| | c.1851_1852delTA | p.Asp617GlufsTer9 | 1 | 0 | 0 | 0 | 5 |
| | c.1087C > T | p.Arg363Ter | 1 | 0 | 0 | 0 | 0 |
| | c.219_220insA | p.Asn75LysfsTer16 | 0 | 0 | 1 | 0 | 0 |

n(cases/controls): Exomes(1006/1,609), WGSET(188/3,582), ExAC(0/27,173)
CRC, colorectal cancer; ExAC, Exome Aggregation Consortium; GSEA, gene set enrichment analysis.
Genes identified by GSEA.

maintenance; in addition, MRE11A is a double-strand break repair nuclease involved in homologous recombination and is inactivated in mismatch repair-deficient cancer. Three cases had disruptive mutations in *MRE11A*, including the ovarian cancer-associated mutation p.Leu7fsTer18 (ref. 11), which was identified in a 55-year-old male with microsatellite stable rectosigmoid adenocarcinoma. There was no documented family history of ovarian cancer (Supplementary Table 4). In *POT1*, we identified three disruptive mutations (two WES and one WG188; Supplementary Table 4) including the familial glioma variant p.Asp617GlufsTer9, predicted to impair association with telomeres[12]. In addition to glioma, inherited *POT1* mutations are documented to cause familial melanoma[13,14], Li-Fraumeni syndrome[15] and chronic lymphocytic leukaemia. The mother of the p.Asp617GlufsTer9 mutation carrier had CRC at 47 and lymphoma at 76 years. Our findings are compatible with a more extensive spectrum of cancer types associated with *POT1* mutation.

**Search for recessively acting variants.** Although most Mendelian CRC genes are dominant, inactivation of MUTYH, one of the three key components of base-excision repair, causes recessive CRC[16]. We looked expressly for evidence of homozygosity or compound heterozygosity for rare damaging variants. Although not highly powered to detect such alleles, we identified compound heterozygosity for the base-excision repair gene *NTHL1* p.Gln90Ter/p.Gln287Ter (Supplementary Table 4), supporting the recent observation of *NTHL1* as a rare cause of recessive CRC[17].

**Analysis of common variants.** Although not the primary purpose of this study, we took the opportunity to search for common (that is, MAF > 5%) SNVs and indels influencing CRC risk—identifying a significant association with *ATF1* c.327C > T (rs1129406; MAF = 0.39, $P = 2.08 \times 10^{-7}$).

**Analysis of non-coding variants.** Although our sequencing primarily targeted exons, a proportion of 'off-target' reads are expected to encompass gene regulatory regions (Methods). Acknowledging the limitations of these data we examined such variants, identifying a significant association with a common variant rs749072 (chr3.37096024:g.T > C, MAF = 0.36, $P = 8.0 \times 10^{-8}$), which is associated with *MLH1* promoter methylation[18]. No rare non-coding variant showed a significant association.

**Discussion**

Here we have searched for high-impact mutations within the exome, a highly enriched subset of the genome in which it has been argued that disease-causing mutations are most likely to reside. By focusing on the exome, we have limited our ability to identify pathogenic mutations outside of transcribed regions and targeted capture is insufficiently sensitive to detected copy number variation. However, data catalogued by CLINVAR on the known CRC genes suggests copy number variations (1–50 kb) are likely to account for <10% of pathogenic mutations. Accepting these caveats, our findings invite several conclusions. We can confidently ascribe 15% of familial CRC cases to rare

variants in established CRC predisposition genes with a maximum estimate of 31%. We also have found evidence for novel CRC predisposition genes, which merit further functional and/or replication analysis. The biological function of the candidate CRC genes we identified file (*MRE11*, *POLE2* and *POT1*) further underscore the centrality of DNA replication/instability as a cause of heritable CRC. It is however unlikely that further major high-penetrance genes with a similar profile to the mismatch repair or *APC* genes exist. Moreover, we did not identify any coding low-frequency alleles (1–5%) with moderate effect.

Over the range of allele frequencies and effect sizes compatible with the established 2.2-fold sibling relative risk of CRC[19], our study had over 80% power to identify a new predisposition gene accounting for >1% of the familial risk (Supplementary Fig.3). Our study thus clarifies the genetic architecture of CRC and probably discounts the existence of further major high-penetrance susceptibility genes. For alleles having a more modest impact on overall CRC burden, sample sizes of 10,000 cases are likely to be required for gene discovery initiatives. The identification of missense driven CRC genes will be hampered by our inability to accurately classify such mutations, acting to dilute the association signal and reducing effective population size. Thus, our analysis highlights a major requirement in cancer gene discovery studies, to improve on the ability to accurately assign pathogenicity to sequence changes. In the meantime, additional evidence such as functional data and/or screens of further cases will be required to prioritize candidate pathogenic coding variants. Furthermore, our ability to identify common risk variants, coupled with the prediction from statistical modelling of genome-wide association study (GWAS) data that 17% of the heritable CRC risk can be ascribed to common variants[20], provides a rationale for continued implementation of GWAS to identify new risk loci. It is notable that the mutations in the known genes are associated with more profound family history of CRC than in those in whom a genetic diagnosis cannot be made (Supplementary Table 2). Thus, from the clinical perspective, gene testing should not detract from the value of a detailed family history to inform screening requirements in patients and families.

In conclusion, from our analysis it is clear that the identification of additional significant CRC gene associations will require very large-scale sequencing in conjunction with careful statistical analysis. However, such efforts should not detract from the analysis of small but highly informative CRC families ascertained on the basis of highly selected phenotypes and which are likely to continue to prove highly effective in gene discovery.

## Methods

**Ethics.** Written informed consent was obtained from all individuals with ethical review board approval (UK National Cancer Research Network Multi-Research Ethics Committee 02/0/097) and the study was conducted in accordance with the declaration of Helsinki.

**Samples and data sets for discovery.** The cases comprised 1,028 unrelated patients (559 male) with CRC, aged ≤55 years at diagnosis (mean age 48.7, s.d. = 6.0), who had at least one first-degree relative with CRC, ascertained between 2003 and 2011 through the NSCCG[3]. All the patients were UK residents and had self-reported European ancestry. Germline DNA was isolated from EDTA-venous bloods using standard methods and picogreen quantified.

The controls comprised 1,644 healthy individuals from the UK 1958BC[21]—974 from the ICR1000 data set (EGAD00001001021)[22] and an additional 670 individuals, all sequenced at The Institute of Cancer Research as per the Discovery cases.

**Whole-exome sequencing.** A Covaris E Series instrument (Covaris Inc., Woburn, MA, USA) was used to fragment 1 µg of DNA from each individual. Illumina's Truseq 62 Mb expanded exome enrichment kit was used to prepare indexed paired-end libraries, as per the manufacturer's instructions (Illumina, San Diego, CA, USA); 2 × 100 bp sequencing was performed using Illumina HiSeq2000 technology.

**Read mapping and variant analysis.** CASAVA software (v.1.8.1, Illumina) was used to extract paired-end fastq files. The alignment of reads to build 37 (hg19) of the human reference genome was performed using Stampy[23] and BWA[24] software. The Genome Analysis Tool Kit (GATKv3) pipeline was implemented according to best practices[25,26]. Analysis was restricted to capture regions defined in the Truseq 62 Mb bed file plus 100 bp padding. The Variant Effect Predictor (VEP; ref. 27) was used to provide annotations on the predicted impact of each variant together with functional classifications from the CONDEL[28] algorithm. We additionally annotated with alignability of 100mers and distance from simple repeats defined by UCSC browser tracks.

**Sample-level quality control.** Using the set of common SNVs defined in dbSNPv138 and performing genotype calling using the Platypus[29] algorithm, we assessed sample quality according to a number of metrics. We identified seven individuals who had non-northern European ancestry, through principal component analysis using EIGENSTRAT[30] software in conjunction with HapMap Project data. In addition, we excluded individuals with high levels of heterozygosity, sex discrepancy, poor call rate and contamination (Supplementary Fig. 1). During the course of the study, two 1958BC controls were identified as having been diagnosed with cancer and they were excluded. We calculated a similarity metric between all samples to assess identity-by-state; no related individuals were identified. Cases and controls were compared using transition/transversion ratios, dbSNP percentage and number of alternate alleles across of different VEP classes called by the GATK pipeline. No substantial difference was observed between these metrics for cases and controls (Supplementary Table 1).

**Variant filters.** We considered only canonical transcripts and for each variant, assuming the most deleterious predicted effect for each transcript according to VEP. In gene-based analyses, only non-silent variants were considered (that is, transcript ablations, splice donor/acceptors, splice region, stop gain, frameshift, stop lost, initiator codon variants, transcript amplifications, in-frame insertion/deletions and missense). For all analyses, we imposed GATK internal calling thresholds excluding variants as per the current best practice guidelines—in the 99.5th truth tranche for SNVs and >99th tranche for indels. To further identify false positives, which would have an adverse impact on the analysis, we adopted an automated approach imposing: GQ ≥30; for a heterozygous call, an alternate depth ≥3 and $\chi^2 < 10.83$ (that is, $P > 0.0001$) for the observed versus expected distribution of alternate/reference alleles (alt-ref-ratio); UCSC alignability (100 bp window size) = 1, not in simple repeat; Hardy–Weinberg equilibrium test ($P > 1.0 \times 10^{-8}$) in cases and controls; and an overall call rate ≥75% in both cases and controls.

**Contamination analysis.** We used verifyBamID to estimate per sample levels of contamination, identifying a single contaminated sample, which was excluded.

**ExomeArray concordance.** We evaluated the fidelity of exome sequencing in 1,332 samples, which had also been genotyped using Illumina HumanExome-12v1_A Beadchip arrays (Illumina). Probes were excluded if monomorphic, call rate <0.99 in cases/controls, there was a significant difference in uncalled genotypes between cases and controls ($P < 0.05$), there was a significant Hardy–Weinberg equilibrium in controls $P < 0.001$ and if non-autosomal. Samples were excluded if the call rate was <0.99 or called variants exhibited outlying heterozygosity (>3 s.d.). Concordance was assessed for rare biallelic SNVs (<5% frequency). SNVs with unresolvable strands (A/T and C/G) were excluded. We used PLINK-seq to extract a set of 46,811 SNVs seen in the exome array data. Assuming that this set constitutes the set of true positives, we assessed the sensitivity of our exome sequencing data by counting the number of these variants correctly identified through our exome-sequencing protocol. Specificity and sensitivity across all alleles with MAF <0.05 was >99.99% and 78.4%, respectively, for filtered variants.

**Coverage of non-coding regions.** The Illumina expanded TruSeq Exome Enrichment captures 62 Mb of the human exonic regions, including 28 Mb of 5′- and 3′-untranslated regions. As part of the analysis pipeline, the TruSeq capture regions are padded by an additional 100 bp at either end as per GATK best practices. In total, 46 Mb of intronic sequence and 21 Mb of promoter regions are captured (62% at >10× coverage).

**Samples and data sets for replication.** The WG188 cases comprised 188 CRC cases selected for early age of presentation and family history from the Colorectal Tumour Gene Identification study (http://public.ukcrn.org.uk/search/StudyDetail.aspx?StudyID=7590). All of the cases were UK residents and had self-reported European ancestry. All were mutation negative for mismatch repair mutations. WGS data (mean average coverage 58×) were obtained using the Complete Genomics technology[31]. Variant call format files were extracted using CGAtools. Only exonic variants were considered for this analysis.

UK10K individual-level variant call format files were downloaded from the Sanger ftp site. ALSPAC (1,828) and TWINSUK (1,754) samples that passed

UK10K quality control metrics were extracted. Only coding variants that had passed UK10K quality control were considered. European population frequencies of coding variants were downloaded from ExAC—release 0.3. For this analysis, a subsection of the ExAC data was used, excluding samples analysed as part of The Cancer Genome Atlas.

The Illumina Exome array data comprised Infinium Human Exome BeadChip 12v1.0 or 12v1.1 exon array data on 5,552 UK cases and 6,792 UK controls, which we have previously reported[32]. We excluded samples overlapping with the Discovery exome-sequencing set.

**Statistical analysis.** For individual variant association, we performed a per-variant analysis (considering each alternate allele in turn) using a two-sided Fisher's exact test implemented in R software (version 3.1.3). Meta-analysis was performed with exome array data sets to combine evidence across study-specific P-values under a fixed-effects model.

For gene-centric analysis, to test whether rare mutations contribute to CRC, we performed a collapsing burden test imposing a maximal MAF threshold of 1% (T1 test). We applied the T1 test to three different types of variant groups: (1) Disruptive (stop gain and frameshift); (2) Predicted damaging (stop gain, frameshift, missense predicted to be damaging by CONDEL and splice site acceptor/donors); and (3) Non-synonymous (all coding non-synonymous variants). Significance levels were assessed using $10^5$ permutations on case/control status. Genes with a T1 $P < 0.01$ were permuted an additional $2 \times 10^6$ times, to ensure recovery of statistical significance for associations. Exome-wide significance was considered to be $P = 8.0 \times 10^{-7}$, corresponding to a Bonferroni correction for the testing of $\sim 20,000$ genes and three variant sets.

The T1 burden test was applied to both Discovery (exome) and WGSET sets. For the Discovery set, the T1 test was performed on both the full set of 1,006 cases and a set of cases where CRC could not be confidently ascribed to pathogenic variant in an established CRC predisposition gene. To create this reduced set, we took a conservative criterion to maximize the opportunity of discovering a novel association. Those samples that contained either a Class 1 variant or a variant previously described as being pathogenic or likely to be pathogenic by InSight (The International Society for Gastrointestinal Hereditary Tumours) were removed, leaving 863 samples potentially harbouring novel associations.

The replication analysis, based on the WGSET, was restricted to genes with Class 1 nominally significant T1 P-values in this reduced Discovery series. Meta-analysis was performed on the two data sets, to combine evidence across study-specific P-values using sample size-weighted Z-score method, implemented in METAL[33,34].

We assessed the biological plausibility of candidate genes on the basis of interaction with a known high-penetrance gene, being part of a known CRC pathway, mapping to a GWAS signal, being significantly somatically mutated in CRC, established tumour suppressor or oncogenic role.

We used GSEA to assess the potential over-representation of damaging variants within a curated set. Ranking of genes was based their T1 permuted P-values. A pre-ranked GSEA was then performed using sets defined as being part of the same GO biological process (c5.bp.v2.5.symbols.gmt provided by GSEA software). Set-based permutations (10,000) were performed to determine significance. Leading edge analysis was performed on all gene sets stipulating a $Q < 0.25$, to identify genes that accounted for each set's enrichment signal (Supplementary Table 5).

**Calculation of study power.** Disease allele frequency in controls was taken as the baseline allele frequency, whereas the frequency in cases was determined by a weighted average of the enrichment found in cases with one, two and three affected first-degree relatives. Allele counts were then sampled from frequencies between 0.00001 and 0.01, and relative risks between 1.75 and 4.0. A Fisher's test was then performed for each sampling of cases and controls. This process was performed 10,000 times for each frequency/relative risk combination and for each instance the frequency of tests that were significant at an exome-wide significance of $8.0 \times 10^{-7}$ equated to study power.

**Data availability.** WES data on the 1,006 CRC cases and 648 controls have been deposited at the European Genome-phenome Archive, which is hosted by the European Bioinformatics Institute, accession numbers EGAS00001001666 and EGAS00001001667, respectively. The remaining data are either contained within the Article and its Supplementary Information or available from the authors on request.

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

## Acknowledgements

This study makes use of the ICR1000 UK exome series data generated by Professor Nazneen Rahman's team at The Institute of Cancer Research, London. At the Institute for Cancer Research (ICR), the work was funded by a Cancer Research UK grant

(C1298/A8362) and the European Union Seventh Framework Programme (FP7/207–2013) under grant 258236, FP7 collaborative project SYSCOL. B.K. was supported by an ICR PhD studentship funded by the Sir John Fisher Foundation. In Oxford, the work was funded by the Oxford Comprehensive Biomedical Research Centre core infrastructure support to the Wellcome Trust Centre for Human Genetics, Oxford (Wellcome Trust 090532/Z/09/Z). We are grateful to colleagues within UK Clinical Genetics Departments (Colorectal Tumour Gene Identification) and those from the UK National Cancer Research Network (for NSCCG). In Scotland, the work was funded by a Cancer Research UK (C348/A12076) and Medical Research Council Grant (MR/KO18647/1). We are grateful to all members of UK exome array consortium for sharing information on allele frequencies (http://diagram-consortium.org/uk-exome-chip/). In Leeds, the work was funded by Cancer Research UK Programme Grant (C588/A19167). This publication is supported by COST Action BM1206.

## Author contributions

R.S.H., D.C., P.B., B.K., S.E.D., I.T. and M.G.D. contributed to writing of the manuscript. R.S.H., D.C., P.B., B.K., I.T., S.E.D. and M.G.D. conceived and designed the experiments. D.C., M.N.T., P.B., B.K., I.T., S.E.D., M.F., Y.P.M., A.L.S., M.G.D. and R.S.H. performed the experiments. D.C., P.B., B.K., S.E.D., M.F., A.P., A.L.S., Y.P.M. and R.S.H. analysed the data. S.P. and C.P. were involved in study design/sampling/assembly/data collection, collation, curation, sequencing and quality control/data analysis. All authors reviewed the manuscript.

## Additional information

**Accession codes**: The WES data have been deposited at the European Genome-phenome Archive, which is hosted by the European Bioinformatics Institute, under accession codes EGAS00001001666 and EGAS00001001667.

**Competing financial interests**: The authors declare no competing financial interests.

**How to cite this article**: Chubb, D. *et al.* Rare disruptive mutations and their contribution to the heritable risk of colorectal cancer. *Nat. Commun.* **7**:11883 doi: 10.1038/ncomms11883 (2016).

