## [Peer Review File · Nature Communications]

Reviewer #1 (Remarks to the Author): Expert in familial colorectal cancer genetics

A. Summary of the key results

In this manuscript by Chubb et al. the authors present the study of 1,006 exomes from early-onset (<55y)CRC patients with at least one first-degree relative affected with CRC and of 1,609 controls, with the aim of knowing the contribution to the heritable risk of CRC of rare disruptive variants, either in already known hereditary CRC genes or in novel genes. Their findings show that 15-31% of all familial cases are due to rare germline mutations in known high-risk genes. No additional major high-risk genes are identified. Based on different strategies, they propose additional novel CRC genes such as IL12RB1, LIMK2, POLE2, POT1 and MRE11A. They also search for common variants influencing CRC risk.

B. Originality and interest

The study presented in this manuscript is of great interest to other researchers in the field. There are no other studies reported in the literature where such high number of familial CRC cases have been screened by exome sequencing, thus being a great opportunity to assess the real burden of high-risk mutations in hereditary CRC and to identify novel CRC-predisposing genes. Moreover, the inclusion of data from controls makes the study even more robust and unique.

Nevertheless, their findings are not very impressive. The fact that not additional major CRC genes were identified is something we have known for some time now, due to the failure of large linkage studies to identify them. For this reason, I believe the authors should have focused and further studied the novel candidates they identify, even though their prevalence is not as high as that of MMR genes or APC.

C. Data & methodology: validity of approach, quality of data, quality of presentation

The approach is valid, logic and highly intuitive. The quality of data is excellent. Regarding the presentation, I find that the discussion is somehow poor. Most of the interesting findings are not discussed, beginning with the novel potential familial CRC genes they identify. Also, I find the last two paragraphs of the Results section not so easy to understand, especially the second part of the first one (it looks more as something one would find in the discussion section), and the lack of information about which part of the non-coding regions they were able to analyze.

D. Appropriate use of statistics and treatment of uncertainties: yes

E. Conclusions: robustness, validity, reliability.

The conclusions are robust and match their findings.

F. Suggested improvements: experiments, data for possible revision

The manuscript is highly descriptive, which is fine, but I believe that the authors should have studied deeper the novel candidate genes. I miss a deeper description of the mutations found, the carrier families (I do not know if they have access to this information, maybe from the validation series?). An extreme example is for IL12RB1, LIMK2, which are merely once mentioned.

Even if the prevalence of the mutations found in the candidate genes proposed is not so high, they may be relevant for the routine diagnostics of familial cancer, as happens with PMS2, POLE or POLD1. In this particular study mutations in these genes are rarely found and in some instances, they showed the same frequency in cases and controls. Of course, this shows their limited contribution to CRC susceptibility in general, but they are nevertheless important and relevant for routine genetic diagnostics.

G. References: appropriate credit to previous work? Yes

Reviewer #2 (Remarks to the Author): Expert in sequencing

Chubb et al., describe the largest survey to date to identify rare high-penetrance, susceptibility genes variants which contribute to the heritable risk of CRC. They performed whole exome sequencing of 1,006 early onset colorectal patients (and 1,609 controls) and supplemented this with meta-analysis on an additional ~5,000 case and 5,000 controls.

They identify few new high-penetrance, susceptibility genes variants (6 cases with mutations in IL12RB1; 7 case with LIMK2; 3 case with MRE11A; 5 with POLE2; 3 case with POT1) and ascribe ~15% of familial CRC cases to rare variants in established CRC predisposition genes with a maximum estimate of 31%. They conclude that the existence of further major high-penetrance susceptibility genes can be discounted.

The paper is very clear and well written. I would recommend publication with minor changes.

This is an important paper for it provides quantitation for the likelihood that colon cancer is inherited.

Suggested Improvements:

The following important changes should be considered by the authors.

1. Eliminate from the word discounts from abstract and replace it by unlikely.
2. When referring to rare allele make sure it's clear that it is genomic inherited mutations that you are addressing. This is done correctly in the discussion first paragraph but not in the introduction.
3. The question of multiple inherited mutations should be addressed.
4. While the authors recognize the limitations of exome sequencing (i.e., not sampling regulatory regions and bioinformatic prediction of functional consequence of a particular variant), they inadequately stress the possible role large copy number variants could play (and which are largely missed by the current analysis). CNVs account for up to 10% of autism cases (Sebat et al., Science 2007) and up to 20% of young onset cases of schizophrenia (Walsh et al Science 2008), for example. In their discussion of limitations, detection of large structural variants should be more fully discussed.

Consistent with this, it would be the reviewers preference that the adverb 'likely' be included in the abstract's conclusion: 'Our study clarifies the genetic architecture of CRC, and likely discounts the existence of further major high-penetrance susceptibility genes which individually account for > 1% of the familial risk.'
5. The centrality of DNA replication/genetic instability to the known and the here newly identified variants could be further emphasized. This appears to be the major lesson about the biology behind the genetics of those hereditary cases of CRC that can be accounted for so far.

6. As the authors emphasize, it is most striking that >70% of early onset case (likely to have a strong hereditary component) cannot be explained by rare high-penetrance, susceptibility genes variants. Supplementary Figure 3 visually explains the amount of 'missing' inheritance not captured in these analyses even when criteria are relaxed. This figure very emphatically makes the point that at the current time family history is more informative than detailed genomic analyses. As well as capturing in the underlying complexity of the genetics, it also reinforces the clinical value of detailed family histories in patient management. Including Supplementary figure 3 in the main article may therefore be worthwhile, as would mention of the clinical value of detail family history in the concluding paragraph.

REVIEWERS' COMMENTS:

Reviewer #1 (Remarks to the Author):

A. Summary of the key results

In this manuscript by Chubb et al. the authors present the study of 1,006 exomes from early-onset (<55y) CRC patients with at least one first-degree relative affected with CRC and of 1,609 controls, with the aim of knowing the contribution to the heritable risk of CRC of rare disruptive variants, either in already known hereditary CRC genes or in novel genes. Their findings show that 15-31% of all familial cases are due to rare germline mutations in known high-risk genes. No additional major high-risk genes are identified. Based on different strategies, they propose additional novel CRC genes such as IL12RB1, LIMK2, POLE2, POT1 and MRE11A. They also search for common variants influencing CRC risk.

Response: No response required

B. Originality and interest

The study presented in this manuscript is of great interest to other researchers in the field. There are no other studies reported in the literature where such high number of familial CRC cases have been screened by exome sequencing, thus being a great opportunity to assess the real burden of high-risk mutations in hereditary CRC and to identify novel CRC-predisposing genes. Moreover, the inclusion of data from controls makes the study even more robust and unique.

Nevertheless, their findings are not very impressive. The fact that not additional major CRC genes were identified is something we have known for some time now, due to the failure of large linkage studies to identify them. For this reason, I believe the authors should have focused and further studied the novel candidates they identify, even though their prevalence is not as high as that of MMR genes or APC.

Response: We appreciated that the reviewer found our paper of interest.

C. Data & methodology: validity of approach, quality of data, quality of presentation

The approach is valid, logic and highly intuitive. The quality of data is excellent. Regarding the presentation, I find that the discussion is somehow poor. Most of the interesting findings are not discussed, beginning with the novel potential familial CRC genes they identify. Also, I find the last two paragraphs of the Results section not so easy to understand, especially the second part of the first one (it looks more as something one would find in the discussion section), and the lack of information about which part of the non-coding regions they were able to analyze.

Response: We have revised the last two paragraphs of the results moving text to the discussion where appropriate. We have also expanded and revised the discussion. We now provide detailed information about the non-coding regions that we were able to analyse. Specifically we now state in a new section in the Methods entitled "Coverage of non-coding regions" : *The Illumina expanded TruSeq Exome Enrichment captures 62 Mb of the human exonic regions, including 28Mb of 5' and 3' UTRs. As part of the analysis pipeline, the TruSeq capture regions are padded by an additional 100bps at either end as per GATK best practices. In total 46Mb of intronic sequence and 21Mb of promotor regions are captured (62% at >10x coverage).*

D. Appropriate use of statistics and treatment of uncertainties: yes

Response: No response required.

E. Conclusions: robustness, validity, reliability.

The conclusions are robust and match their findings.

Response: No response required.

F. Suggested improvements: experiments, data for possible revision

The manuscript is highly descriptive, which is fine, but I believe that the authors should have studied deeper the novel candidate genes. I miss a deeper description of the mutations found, the carrier families (I do not know if they have access to this information, maybe from the validation series?). An extreme example is for *IL12RB1*, *LIMK2*, which are merely once mentioned.

Response: We have expanded our results section to include a detailed description of *IL12RB1* and *LIMK2* mutations. For these and all novel candidate genes we now provide details of relevant family history.

Even if the prevalence of the mutations found in the candidate genes proposed is not so high, they may be relevant for the routine diagnostics of familial cancer, as happens with *PMS2*, *POLE* or *POLD1*. In this particular study mutations in these genes are rarely found and in some instances, they showed the same frequency in cases and controls. Of course, this shows their limited contribution to CRC susceptibility in general, but they are nevertheless important and relevant for routine genetic diagnostics.

Response: No response required

G. References: appropriate credit to previous work? Yes

Response: No response required

Reviewer #2 (Remarks to the Author): Expert in sequencing

Chubb et al., describe the largest survey to date to identify rare high-penetrance, susceptibility genes variants which contribute to the heritable risk of CRC. They performed whole exome sequencing of 1,006 early onset colorectal patients (and 1,609 controls) and supplemented this with meta-analysis on an additional ~5,000 case and 5,000 controls.

They identify few new high-penetrance, susceptibility genes variants (6 cases with mutations in IL12RB1; 7 case with LIMK2; 3 case with MRE11A; 5 with POLE2; 3 case with POT1) and ascribe ~15% of familial CRC cases to rare variants in established CRC predisposition genes with a maximum estimate of 31%. They conclude that the existence of further major high-penetrance susceptibility genes can be discounted.

The paper is very clear and well written. I would recommend publication with minor changes.

This is an important paper for it provides quantitation for the likelihood that colon cancer is inherited.

Response: We appreciated that the reviewer found our paper of interest.

Suggested Improvements:

The following important changes should be considered by the authors.

1. Eliminate from the word discounts from abstract and replace it by unlikely.

Response: We acknowledge this point (and in conjunction with point 4 below) and have revised our text to read "likely discounts".

2. When referring to rare allele make sure it's clear that it is genomic inherited mutations that you are addressing. This is done correctly in the discussion first paragraph but not in the introduction.

Response: We have revised the introduction to emphasise that we are seeking to identify rare germline predisposition variants for CRC. Specifically, in the first paragraph of the introduction we now state "*The genetic architecture of CRC susceptibility encompasses a broad spectrum of risk; from rare highly penetrant germline mutations associated with well characterised syndromes to common polymorphisms each individually conferring small risks.*". The second paragraph now begins "*The high baseline rate of rare, neutral germline variants makes the identification of rare CRC predisposition alleles problematic*".

3. The question of multiple inherited mutations should be addressed.

Response: We have now included a supplementary note and related tables with details of the number of multiple mutations in cases and controls and an analysis of cases with multiple mutations, focussing on the known genes. The note is referenced from the main text where we now state "*None of the cases were identified as being carrier of multiple Class 1 mutations in a known gene (Supplementary Note)*".

4. While the authors recognize the limitations of exome sequencing (i.e., not sampling regulatory regions and bioinformatic prediction of functional consequence of a particular variant), they

inadequately stress the possible role large copy number variants could play (and which are largely missed by the current analysis). CNVs account for up to 10% of autism cases (Sebat et al., Science 2007) and up to 20% of young onset cases of schizophrenia (Walsh et al Science 2008), for example. In their discussion of limitations, detection of large structural variants should be more fully discussed.

Response: We acknowledge the reviewers comment and as part of the discussion now highlight the possible impact of copy number variation to CRC predisposition acknowledging the limitation of exome sequencing to interrogate this. Specifically we now state *“Here we have searched for high-impact mutations within the exome; a highly-enriched subset of the genome in which it has been argued that disease-causing mutations are most likely to reside. By focusing on the exome we have limited our ability to identify pathogenic mutations outside of transcribed regions and targeted capture is insufficiently sensitive to detected copy number variation (CNV). However, data catalogued by CLINVAR on the known CRC genes suggests CNVs (1-50kb) are likely to account for <10% of pathogenic mutations”*

Consistent with this, it would be the reviewers preference that the adverb 'likely' be included in the abstract's conclusion: 'Our study clarifies the genetic architecture of CRC, and likely discounts the existence of further major high-penetrance susceptibility genes which individually account for > 1% of the familial risk.'

Response: We acknowledge this point (and in conjunction with point 1 above) and have revised our text to read “likely discounts”.

5. The centrality of DNA replication/genetic instability to the known and the here newly identified variants could be further emphasized. This appears to be the major lesson about the biology behind the genetics of those hereditary cases of CRC that can be accounted for so far.

Response: We have revised the discussion accordingly and now state *“The biological function of the candidate CRC genes we identified (MRE11, POLE2, POT1) further underscore the centrality of DNA replication/instability as a cause of heritable CRC”*.

6. As the authors emphasize, it is most striking that >70% of early onset case (likely to have a strong hereditary component) cannot be explained by rare high-penetrance, susceptibility genes variants. Supplementary Figure 3 visually explains the amount of 'missing' inheritance not captured in these analyses even when criteria are relaxed. This figure very emphatically makes the point that at the current time family history is more informative than detailed genomic analyses. As well as capturing in the underlying complexity of the genetics, it also reinforces the clinical value of detailed family histories in patient management. Including Supplementary figure 3 in the main article may therefore be worthwhile, as would mention of the clinical value of detail family history in the concluding paragraph.

Response: We now include Supplementary Figure 3 in the main article and have revised the discussion to comment on the clinical value of recording a detailed family history. Specifically the discussion now reads *“It is notable that the mutations in the known genes are associated with more profound family history of CRC than in those in whom a genetic diagnosis cannot be made (Supplementary Table 2). Thus, from the clinical perspective, gene testing should not detract from the value of a detailed family history to inform screening requirements in patients and families.”*